# Measuring competing outcomes of a single-molecule reaction reveals classical Arrhenius chemical kinetics

Pieter J. Keenan [1,2,3,5], Rebecca M. Purkiss[1,5], Tillmann Klamroth [4], Peter A. Sloan [1,2] & Kristina R. Rusimova [1,2,3] ✉

Programming matter one molecule at a time is a long-standing goal in nanoscience. The atomic resolution of a scanning tunnelling microscope (STM) can give control over the probability of inducing single-outcome single-molecule reactions. Here we show it is possible to measure and influence the outcome of a single-molecule reaction with multiple competing outcomes. By precise injection of electrons from an STM tip, toluene molecules are induced to react with two outcomes: switching to an adjacent site or desorption. Within a voltage range set by the electronic structure of the molecule-surface system, we see that the branching ratio between these two outcomes is dependent on the excess energy the exciting electron carries. Using known values, ab initio DFT calculations and empirical models, we conclude that this excess energy leads to a heating of a common intermediate physisorbed state and gives control over the two outcomes via their energy barriers and prefactors.

The ability to programme matter on the scale of individual atoms or molecules is the pinnacle goal of nanotechnology. Chemical reactions, both desirable (like in the manufacturing of pharmaceuticals) and undesirable (like the below-threshold breaking of DNA strands[1]), are mediated through charge transfer processes which take place on the single-molecule scale. A scanning tunnelling microscope (STM) can initiate and probe these processes in atomically controlled environments. The STM tip is now almost routinely used to control the arrangement of individual atoms and molecules by employing the mechanical interaction between the tip and the target adsorbate[2]. The electric field in the gap between the tip and the sample has also been employed to achieve control over the yield of chemical bond-formation[3] and over the configurational isomerisation of molecules[4]. On the other hand, the STM tunnelling current can, through charge capture or charge-state manipulation, give enhanced specificity and control over a more diverse range of molecular reaction outcomes. On-demand configurational switching of weakly-bound, cryogenically-stabilised single molecules has been demonstrated in many systems[5–7]. Even tip-controlled molecular motors driven by intramolecular proton

transfer and capable of transporting single-molecules have recently been demonstrated[8]. Control over the reaction outcome is typically ensured by exciting different ionic resonances of a molecule[9–12], changing the charge injection location[13–15], or exciting different vibrations[16,17]. In all of these examples, the reaction outcome is altered by changing the initial excitation mechanism. The ensuing dynamics are allowed to evolve naturally from this point but with only one possible reaction outcome.

Here, within an excitation energy window, we keep both the initial conditions and the excitation process of a test system the same: same injection site location, same electronic state excitation. We show that above an energy threshold of +2.0 V, manipulation is mediated by the surface with a state-specific manipulation probability and branching ratio, resulting from an ultrafast energy relaxation of the injected electron. Below that energy threshold, it is the properties of the transient ionic excited state that govern the outcome probabilities of competing chemical reactions, following a similar energy relaxation which takes place within the molecular LUMO state. By employing classical chemical kinetics, we suggest a model whereby the excess

[1]Department of Physics, University of Bath, Bath, UK. [2]Centre for Nanoscience and Nanotechnology, University of Bath, Bath, UK. [3]Centre for Photonics and Photonic Materials, University of Bath, Bath, UK. [4]Universität Potsdam, Institut für Chemie, Theoretische Chemie, Potsdam, Germany. [5]These authors contributed equally: Pieter J. Keenan, Rebecca M. Purkiss. ✉e-mail: k.r.rusimova@bath.ac.uk

energy of the excited state leads to a heating of the intermediate physisorbed state, giving control over the reaction outcomes through their differing energy barriers and Arrhenius prefactors.

## Results

### Single-molecule manipulation with multiple reaction outcomes

Single toluene molecules adsorb onto the Si(111)−7 × 7 surface by attaching to an adjacent adatom-restatom pair on the surface in a 'butterfly' configuration[18]. They form two covalent chemical bonds with the silicon atoms (2, 5-di-$\sigma$ bonding), which are strong enough for stable imaging with a scanning tunnelling microscope at room temperature[19]. Using passive imaging parameters (+1 V and 100 pA), the toluene molecules appear as dark features in Fig. 1a at the regular locations where we expect bright adatoms via tunnelling through the adatom empty-state orbitals. That route for tunnelling electrons is blocked since the 2, 5-di-$\sigma$ bonding saturates the dangling bonds of the underlying silicon adatom-restatom pair (see Fig. 1c and blue and pink circles on the corresponding schematic in Fig. 1d). Figure 1b shows the undoped silicon surface where all remaining dark features are either the periodic corner hole locations of the Si(111)−7 × 7 unit cell (in pink) or adatom vacancies. The toluene-silicon bonds can be controllably cleaved by increasing the energy of the tunnelling electrons above an energy threshold (1.4 eV) defined by the electronic structure of the molecule[20,21] which is well below the carbon-methyl dissociation energy (4.4 eV)[22]. We find the toluene molecule can either switch to a (free) adjacent adatom site or move further (either complete desorption from the surface or longer range diffusion). Here, we refer to the latter as 'desorption'. The outcome of such single-molecule reactions, desorption or switching, can be established by subsequent imaging of the same area. But herein lies the challenge—a chemical reaction takes tens of femtoseconds to occur; in scanning tunnelling microscope

(STM) experiments, an electron arrives at the molecule every picosecond or so; and the time resolution of STM electronics is typically milliseconds. This stark disconnect between the timescale of chemical reactions and the time resolution of the experimental technique means that we can only infer the mechanism of single-molecule reactions from a measurement of the probability of manipulation rather than a time-resolved measurement. Moreover, contained within that reaction probability are other factors: the local density of electronic states of the sample, the electronic configuration of the STM tip at the time of the experiment, thermal and electric-field contributions. It is therefore difficult to deconvolve the measured reaction probability in order to determine the mechanisms behind multiple reaction outcomes that would allow reaction control.

Figure 1c, d shows the three possible outcomes that we measure in our charge-injection experiments: (I) desorption, where the molecule has detached from both its original adatom and restatom, revealing a clean (bright) silicon adatom (II) switching, where the molecule has moved to an adjacent adatom site, seemingly retaining or reforming its original restatom bond[23,24], and (III) no reaction, where the manipulation site appears unaltered in the subsequent STM image. The number of unreacted molecules is included in our analysis of the probability of manipulation but is explicitly excluded from the branching ratio analysis. Instead, we specifically chose the two manipulation outcomes— desorption and switching—as both should have a common initialisation through a physisorbed state[25,26], with switching having the lowest activation energy barrier, and desorption (either complete detachment or diffusion to a non-adjacent adatom site) a higher barrier[19]. Thus the ratio of these two outcomes should be sensitive to any change in the common excited state dynamics. The scenarios where the molecule does not react or reattaches to its original adsorption site, following an excitation into the physisorbed state, do not change

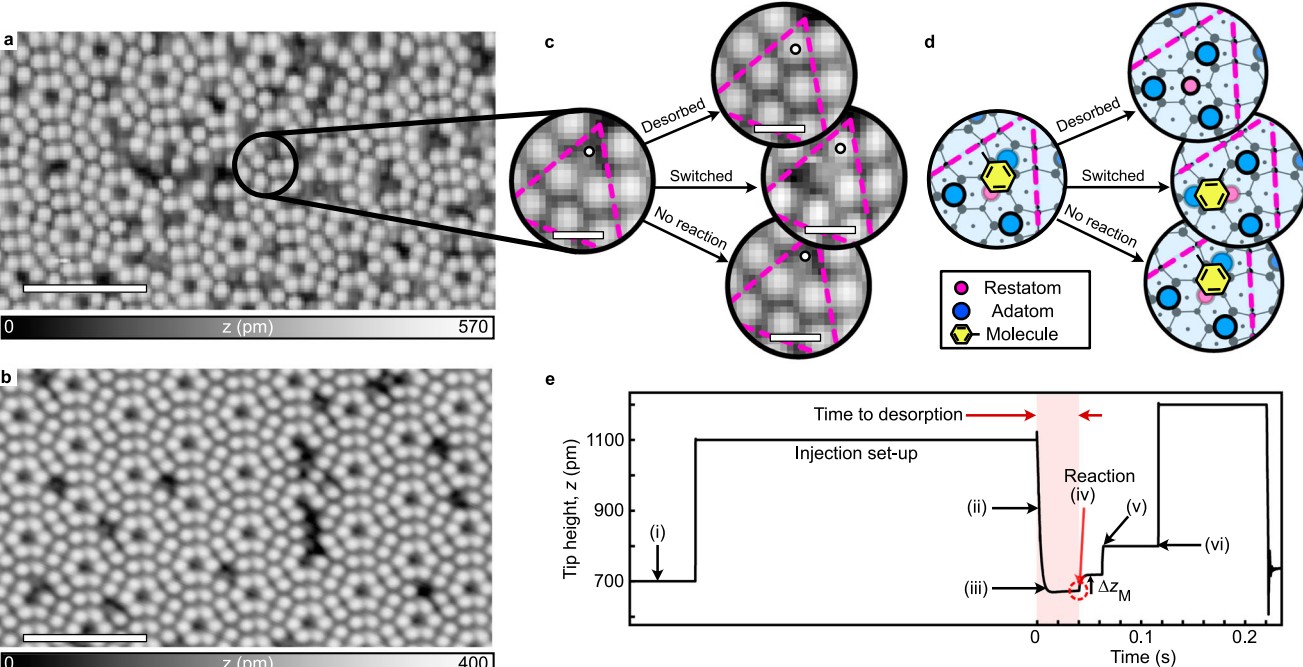

**Fig. 1 | Inelastic tunnelling electron-induced single-molecule manipulation.** 19 nm × 11 nm images (+1.0 V, 100 pA) of the (**a**) Si(111)−7 × 7 surface dosed with 1 L of toluene molecules and (**b**) undosed Si(111)−7 × 7 surface. Scale bars: 5 nm. **c** STM images of a single toluene molecule before (left) and after (right) electron injection at +1.6 V and 300 pA into white circle (imaging parameters: +1 V, 100 pA, 4 nm × 4 nm). STM images on the right show the three possible outcomes: desorption (top), switching (middle), and no manipulation (bottom). Pink dotted triangle indicates a half-unit cell of the Si(111)−7 × 7 surface. Scale bars: 1 nm.

**d** Corresponding schematic diagrams of the STM images in a) showing the locations of adatoms (blue circles), restatoms (pink circles) and molecule (yellow hexagon). Pink dotted triangle indicates a half-unit cell of the Si(111)−7 × 7 surface. **e** Tip height recorded during a charge injection at −1.2 V and 300 pA. Highlighted in red is the duration of the manipulation experiment. The dashed red circle indicates the moment when the molecule reacts. Source data are provided as a Source Data file.

the overall behaviour of the measured outcome branching ratios but just the overall values. Therefore, they are deliberately excluded from our analysis of the reaction outcome branching ratio (see Supplementary Note 4 for a detailed reaction outcome probability tree).

We choose the branching ratio between the two outcomes as our measurement metric, rather than the raw probabilities. As long as we capture several individual manipulation events with the same tip state, any variation due to the tip state, e.g. the gap **E**-field, small variation in position, or any other factor, will effectively cancel out in the analysis of branching ratios. This is evident when we report the current dependence of manipulation. We do, of course, ensure as much as we can, that those factors are controlled and held constant–using computer control, image cross-correlations, etc.

The negative ion resonance-induced manipulation of the single toluene molecule in Fig. 1c is achieved by halting the STM tip on top of the toluene molecule and 'injecting' electrons with preset tunnelling current parameters (here, +1.6 V and 300 pA) in a similar fashion to ref. 27. During the injection experiment, the height of the probe tip is recorded as shown in Fig. 1e: (i), the tip is halted atop the molecule, still at the passive imaging parameters. To protect the tip state while switching to the desired injection parameters, the feedback loop is disengaged and the tip is retracted 0.4 nm away from the surface; (ii) the feedback loop is re-engaged; (iii) the tip approaches the surface, this time at the height corresponding to the injection parameters; (iv) after 20 ms of electron injection, the molecule reacts: the bond with the underlying silicon adatom is cleaved and the tip retracts by a distance $\Delta z_M$ to compensate for the higher local density of electronic states of the bright silicon adatom. At this point, in an ideal experiment the injection should stop - if any further electrons are injected, the molecule, which may still be in the vicinity of the tip, could continue to be excited and so could react again. Therefore, the subsequent STM image may not be a true picture of a single reaction outcome. To stop the electron injection after the manipulation has taken place, we set a threshold of tip height change of $\Delta z > 30$ pm (see Supplementary Note 1). Once such a change is detected by the control electronics, the experiment is stopped in step (v) by reducing the injection current to a minimum safe value of 5 pA. In step (vi) the feedback loop is disengaged and the STM tip is pulled back again to return to the passive imaging parameters. The time delays between steps (iv) and (v), and (v) and (vi) correspond to the minimum response time of the STM control electronics and are the limit of our present experimental capabilities.

At each injection energy, we induce and record over 100 single-molecule reactions. Assuming a first-order rate equation $dN(t)/dt = -N(t)k_d - N(t)k_s$ for the fraction of manipulated molecules $N$ after a time $t$, we can deduce an overall time-dependent probability of manipulating a single molecule given by $P(t) = 1 - \exp(-kt)$, where the measured overall rate $k$ is the sum of the two outcomes: $k_d$ for the rate of desorption and $k_s$ for the rate of switching. Figure 2a shows this for injections at +1.4 V and +1.9 V at 750 pA. To extract the specific rate of manipulation associated with each reaction outcome, $k_d$ and $k_s$, we measure each population as a function of time. Every time a molecule desorbs or switches, we record the reaction outcome and so we can look at the branching ratio in much the same fashion as a coin-flip experiment. We treat our experiments collectively as if we are following a set of $N_0$ molecules and measuring when they desorb or switch and thus we measure the time-dependent population of those two outcomes, $N_d(t)$ and $N_s(t)$. Combining those measurements with the number yet to react $N(t)$ gives $N_0 = N(t) + N_d(t) + N_s(t)$. Therefore the time-dependent population of switched molecules is $dt_s(t)/dt = N(t)k_s$, and similarly $dN_d(t)/dt = N(t)k_d$. Taking the ratio of these and integrating gives $N_d(t) = (k_d/k_s)N_s(t)$ with the branching ratio defined as $B = k_d/k_s$. Therefore Fig. 2b which shows $N_d(t)$ vs. $N_s(t)$ has a gradient equal to the branching ratio of our single-molecule reaction. In this way, the branching ratio is extracted from a direct measurement of its time dependence, rather than by looking at only the final data

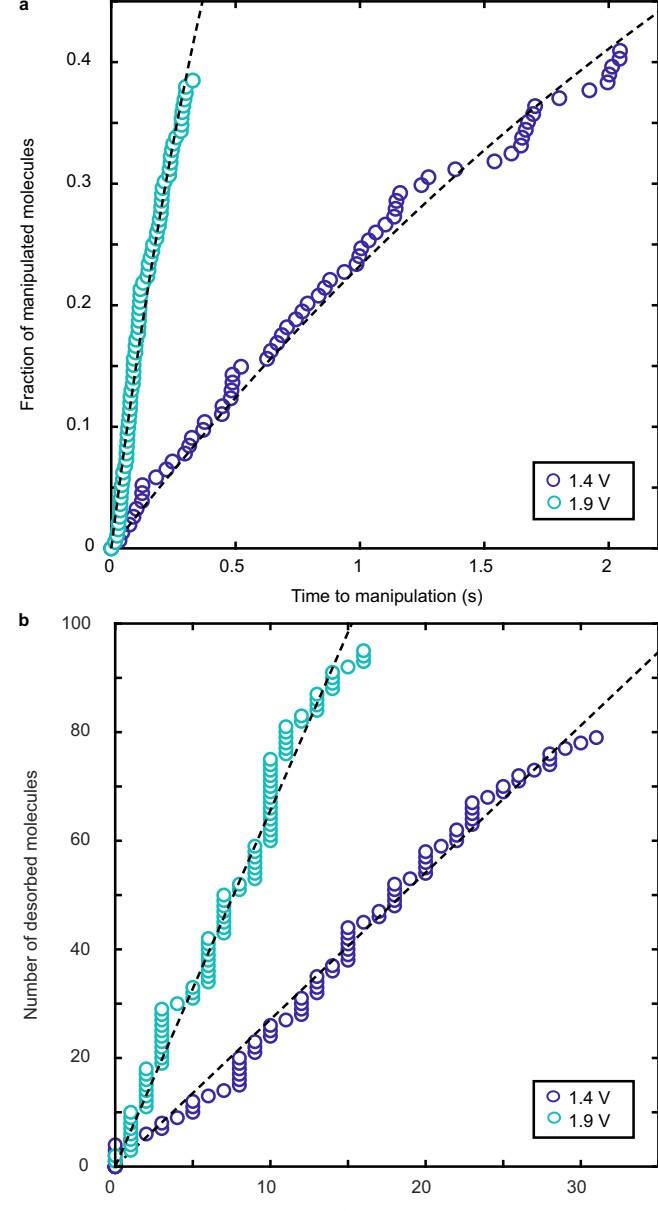

**Fig. 2 | Measured molecular reaction outcomes. a** Time dependence of the fraction of manipulated molecules during injections at the lowest (+1.4 eV, blue circles) and the highest (+1.9 eV, cyan circles) electron energies we used and 750 pA. Each series contains more than 100 individual manipulation events. Dashed lines show the fits to $P(t) = 1 - e^{(-kt)}$ for each injection voltage. **b** Measured time-dependence of the outcome of the single toluene molecule reaction (desorption vs. switching) for the experiments from (**a**). Dashed lines show linear fits to the data. Lateral tip drift speed is kept to below 3 pm s⁻¹ through drift compensation as described in the main text. Source data are provided as a Source Data file.

point. An alternative method for extracting this time-dependent branching ratio involves fitting to each pathway separately. This method is discussed in Supplementary Note 2. Before turning to why the gradient (branching ratio) has a voltage dependence, we need to first probe the manipulation mechanism itself.

## Tunnelling current and injection position dependence

Figure 3a shows the tunnelling current dependence of the desorption and switching rate for electrons injected at +1.6 V. Each data point comes from an analysis of data similar to Fig. 2a. The solid lines

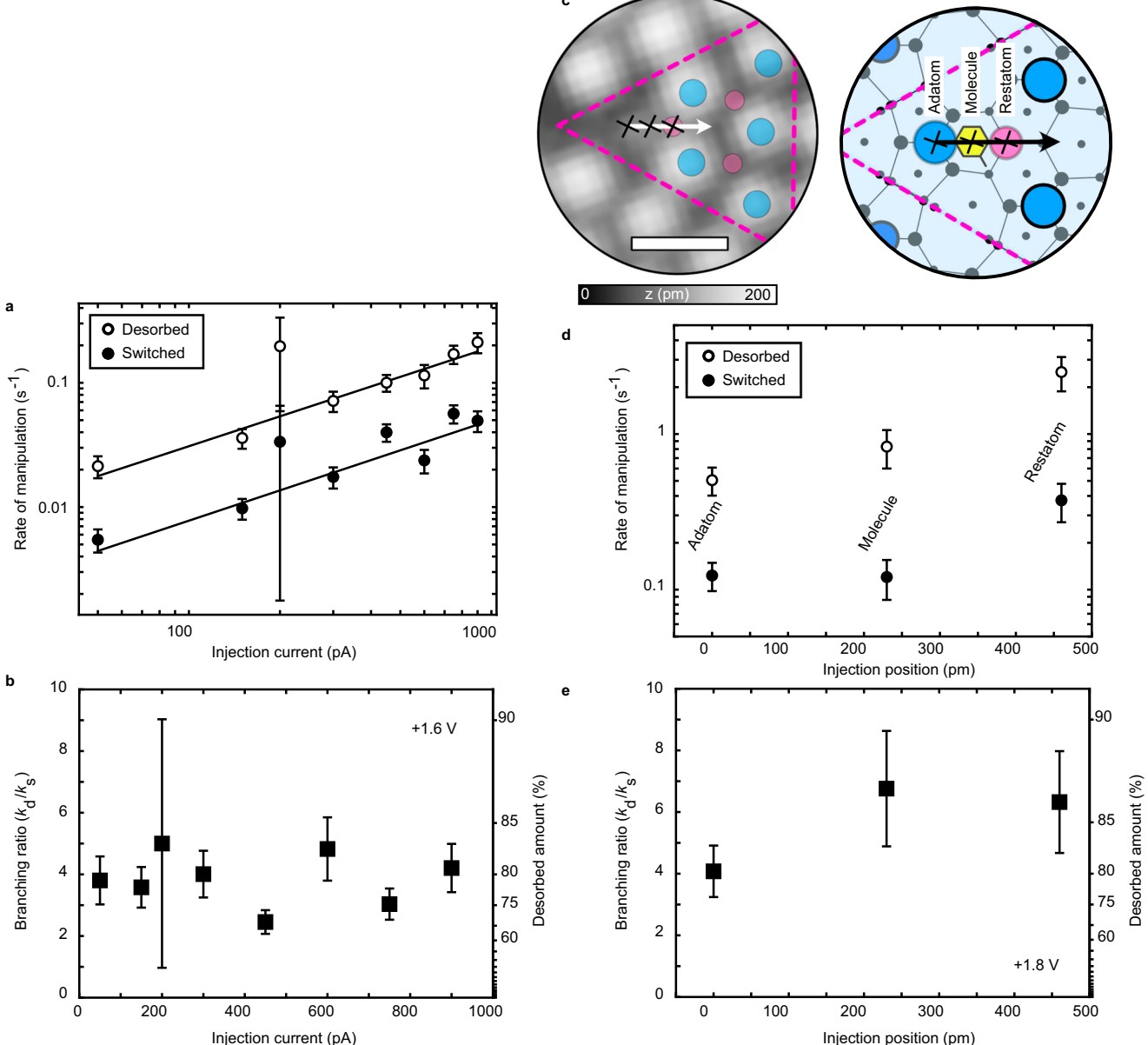

**Fig. 3 | Current and position dependence of single-molecule reaction rate and branching ratio. a** Rates of desorption (unfilled circles) and switching (filled circles) for injections at +1.6 V. At each injection current, we induce and record ≈100 single-molecule reactions, apart from at 200 pA, where only 8 reactions were induced. Error bars indicate the standard deviation. **b** Measured branching ratio between the two active reaction channels (desorption/switching) for the experiments in (**a**). Error bars represent uncertainty from binomial statistics. **c** Smoothed STM image from Fig. 1a) and corresponding schematic with the three different injection locations ((1) adatom (blue circles), (2) molecule (yellow hexagon), and (3) restatom (pink circles)) indicated with black ×. Pink dotted triangle indicates a half-unit cell of the Si(111)−7 × 7 surface. Scale bar: 1 nm. **d** Probability per electron of desorption (unfilled circles) and switching (filled circles) as a function of injection position for injections at +1.8 V and 750 pA. At each injection location, we induce and record ≈100 single-molecule reactions. Error bars indicate the standard deviation. **e** Corresponding branching ratio between the two active reaction channels (desorption/switching) as a function of injection position. Error bars represent uncertainty from binomial statistics. Source data are provided as a Source Data file.

represent power law fits, where the rate, $k \propto I^n$, and where $n$ is the number of electrons that drive a single-molecule reaction. For desorption $n = (0.8 \pm 0.1)$ and for switching $n = (0.8 \pm 0.1)$ which suggests that both outcomes are driven by a one-electron process. Such non-integer results are typical for these experiments due to the sensitivity the raw probabilities can have on the precise location of the tip relative to the target or the tip state. For example, the pioneering early STM reports of hydrogen desorption from Si(100) gave $n = 15$[28] or $n = 10$[29]. However, a following report with noticeably larger current range, reported $n = 0.3, 1.3, 0.3$ depending on the exact experimental procedure, evidence of (on average) a one-electron process[30]. Therefore,

the absolute rates, $k_d$ and $k_s$, will depend on the exact state of the STM tip, whereas the branching ratio of the rates will naturally lead to a normalization of these tip-state effects. Figure 3b shows a near-constant branching ratio over the probed tunnelling current range. This invariance with tunnelling current further shows that both outcomes are one-electron processes and that there is no tip-induced quenching of the excited state[27]. For multi-electron processes, e.g. dynamics induced by multiple electronic transitions (DIMET), the branching ratio can exhibit orders of magnitude difference when the molecule is excited with different tip states or with different currents[10,17]. The uncertainties of Fig. 3b are calculated through purely

binomial statistics. A more accurate error estimate can be obtained through beta distribution analysis[31], which accounts for the finite number of experimental measurements, and yields similar values to the simpler binomial statistics. For all data points, except at 200 pA, the uncertainties represent a fractional uncertainty of ≈20%. To reduce the uncertainty to ≈1%, we would need to perform 62, 500 individual molecular experiments. The data point at 200 pA, where only 8 molecules were manipulated, illustrates clearly the importance of inducing a large number of manipulation events for reliable statistics. To highlight the variation in the branching ratio, the percentage of manipulated molecules that desorb is indicated on the right hand axis in all figures.

Figure 3c–e shows the position dependence of our single-molecule reaction. We inject electrons at +1.8 V and 750 pA into three distinct crystallographic locations of the Si(111)−7 × 7 surface as marked on the STM image and corresponding schematic diagram in Fig. 3c. Location (1) is the centre of the adatom forming a σ bond with the toluene molecule. Location (2) is the mid-point between the adatom and adjacent rest-atom, which we take to be the centre of the adsorbed toluene molecule. Location (3) is on top of the restatom forming the other σ bond with the toluene molecule. For each injection location, we again induce over 100 individual single-molecule reactions. Figure 3d, e shows the position dependence of the manipulation rates and their corresponding branching ratios. The desorption rate is about an order of magnitude lower for injections into (1), the adatom site, compared to the other two injection locations. A similar, yet less striking, trend is observed for the switching rate. By contrast, the branching ratio is largely independent of the injection location, with only a small drop in the measured branching ratio for injections into (1), the adatom site. To ensure that we are always injecting into the same position, and thereby remove this sensitivity to the injection site in the current and voltage sweeps, before each injection we take multiple passive images and use their cross-correlations to measure the lateral tip drift and to feedback a compensation drift value until the measured lateral tip drift speed is below 3 pm s⁻¹.

## Energy dependence

A more pronounced trend is observed for the rate of manipulation and branching ratio as a function of energy of the injected electrons, shown in Figs. 4, 5. The measured combined rate of manipulation (i.e. both desorption and switching) is displayed in Fig. 4d on a linear scale, and 4e on a logarithmic scale. It demonstrates a near-exponential onset at +1.4 V and the measured manipulation rate spans across two decades. In ref. 32 we developed a model to quantitatively link the local density

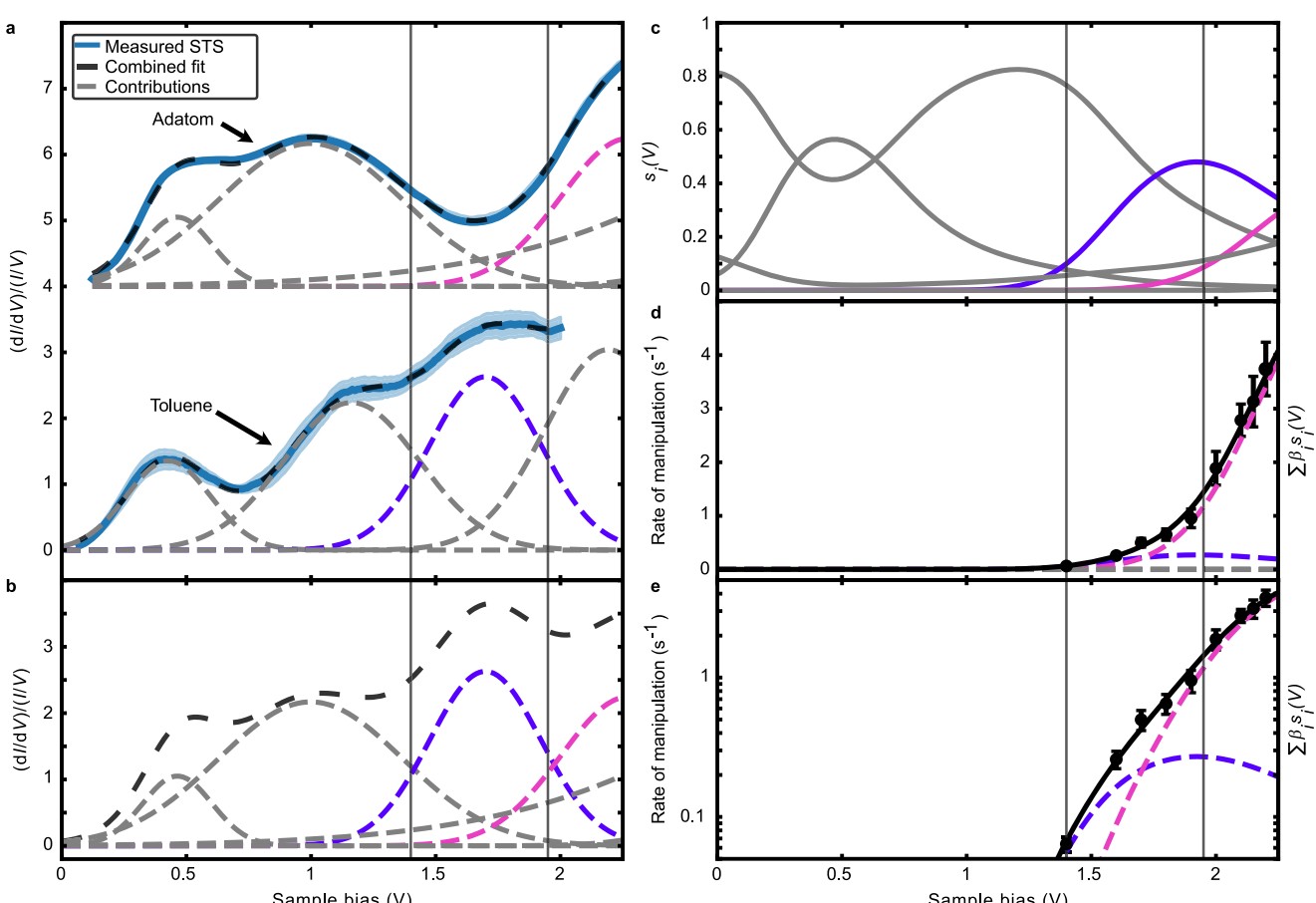

**Fig. 4 | Energy dependence of single-molecule reaction probability for injections into the adatom site. a** Variable gap scanning tunnelling spectroscopy (STS) of top (offset added for clarity): a clean adatom (solid blue line, average of 25 spectra), and bottom: an adsorbate toluene molecule (solid blue line, average of 7 spectra). Shaded error bars represent the standard deviation. The individual Gaussian fits to the STS curve are displayed below each curve, with the combined fit shown with the dashed black line. The two manipulation active states are the U₂ surface band of Si(111)−7 × 7 in pink, with central position at +(2.3 ± 0.3), and the molecule-derived LUMO state shown in blue and centred at +(1.7 ± 0.2) V.

**b** Composite spectrum combining the adatom STS with the molecule-derived LUMO state from (**a**) (see main text for details). **c** Computed fraction $s_i(V)$ of the tunnel current populating each state based on the spectrum in (**b**). Measured rate of manipulation (either desorption or switching) as a function of injection energy on (**d**) a linear scale and (**e**) a log scale. At each injection energy, we induce and record ≈100 single-molecule reactions. Error bars indicate the standard deviation. The fit to the fraction of tunnelling current captured in the energy states from (**b**, **c**) is shown with the solid black lines, together with the individual contributions in dashed lines. Source data are provided as a Source Data file.

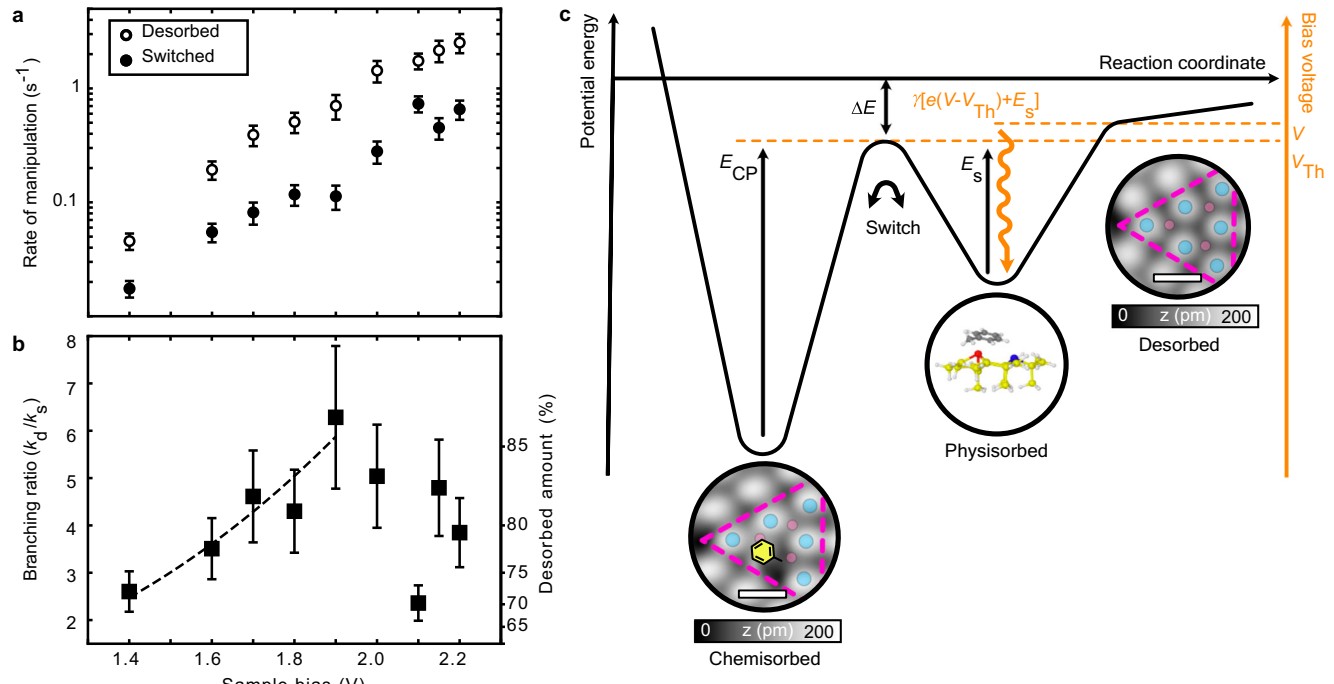

**Fig. 5 | Energy dependence of probability of desorption or switching, and branching ratio for injections into the adatom site. a** Probability of desorption/ switching (unfilled/filled circles) as a function of injection energy. At each injection energy, we induce and record ≈ 100 single-molecule reactions. Error bars indicate the standard deviation. **b** Corresponding branching ratio between the two observable reaction channels (desorption/switching) as a function of injection energy. Dashed line shows fit to the classical thermal model described in the main text.

Error bars represent estimated certainty from binomial statistics. **c** Schematic diagram of the potential energy landscape (black axes) of an adsorbed molecule indicating the energy barriers described in the main text. Right axis (orange) shows the STM injection bias voltage. For simplicity, the route to switching here is presented the same as the route back into the original chemisorbed potential. Scale bars: 1 nm. Source data are provided as a Source Data file.

of states of the surface to the measured reaction probability per injected electron in molecular nanoprobe experiments. In molecular nanoprobe experiments, the STM tip first injects a charge into a surface state. Following an ultrafast relaxation within the surface state, the charge then propagates across the surface causing a molecular manipulation event at a location that is tens of nanometres remote from the original tip position[33,34]. By considering the first step of this nonlocal manipulation effect to be the same as the first step of a scanning tunnelling spectroscopy (STS) measurement we showed that the increase in the rate of manipulation $k$ as a function of energy of the injected charge is in fact due to the overall fraction of the tunnelling current $s_i(V)$ captured by the specific surface electronic state. Therefore, the rate of manipulation $k$ is proportional to the product of the calculated $s_i(V)$ and a state-specific probability of manipulation per injected charge carrier $\beta_i$

$$k \propto \Sigma_i \beta_i s_i(V). \tag{1}$$

Here, we apply this model to the manipulation rate measured by injecting directly into an adsorbate toluene molecule. Figure 4a shows empty-state STS measurements taken over 25 clean silicon adatoms and 7 unreacted toluene molecules. Gaussian functions are fitted to each spectrum and the overall fits show good agreement with the measured data. In the molecular spectrum, we identify the state with a Gaussian fit centred at +1.7 V as the LUMO of toluene (highlighted in blue). It is the onset of this state that is responsible for the observed reaction onset threshold at +1.4 V (indicated by the vertical line). The two lower-lying states are derived from the silicon surface (see refs. 26,35) and are also present in the adatom STS. Any attempt to measure states above +2.0 V results in the immediate manipulation of the

molecule. Instead, for completeness, here we also present the measured STS of a clean silicon adatom, which contains the surface-derived and manipulation-active state, $U_2$[36], with a corresponding Gaussian fit centred at +2.3 V and highlighted in pink. Figure 4b presents a spectrum combining the clean adatom STS with the LUMO state from the toluene STS. Figure 4c shows the computed fraction of tunnelling current, $s_i(V)$, captured by each state calculated from the Gaussian parameters in Fig. 4b as per the recipe from ref. 32. The total rate of manipulation is therefore simply given by the weighted sum in Eq. (1) and results in an excellent fit to the experimental data in Fig. 4d, e. At +1.4 V all of the current that contributes to manipulation is captured by the toluene LUMO state and manipulation is mediated through the molecule, with the manipulation probability per electron given by the only fitting parameter $\beta_{LUMO} = 1.2 \times 10^{-10}$. This is in agreement with previous reports of the energy threshold of electronic manipulation[20,21]. As the bias voltage increases further, the $U_2$ state opens up and competes for the current. It starts to dominate the manipulation above ≈ +2.0 V (vertical line) due to its higher manipulation probability per electron $\beta_{U_2} = 3 \times 10^{-9}$ and results in the nonlocal manipulation process previously reported[33,37]. Importantly, the fit to this model implies an ultrafast energy relaxation of the injected electron prior to manipulation. Below the +2.0 V threshold the manipulation is mediated directly through the LUMO of the molecule. Above this threshold, the electron relaxes down to the bottom of the $U_2$ state before molecular manipulation takes place.

The rates of manipulation for desorption and switching are presented separately in Fig. 5a. Both outcomes display a similar trend to the overall probability of manipulation from Fig. 4, with near-exponential increase in the manipulation rate with the energy of the injected electrons. By contrast, the branching ratio between the two measured outcomes increases monotonically up until a threshold of

+2.0 V. This increase corresponds to a desorbed percentage change from ≈70% to over 85%. Above this bias voltage, the branching ratio is independent of the energy of the injected electrons. This invariance in the measured outcome branching ratio with electron energy is easily explained by considering the ultrafast relaxation step proposed above since all manipulation events take place after the electron has relaxed down to the bottom of the $U_2$ surface state. Therefore, similar to the state-specific probability of manipulation, there is also a state-specific outcome branching ratio set by the energy of the relaxed electron. (The data point at +2.1 V is discussed in more detail in Supplementary Note 3 and the variation is attributed to the quality of the data at this injection bias voltage.)

## Discussion

The increase in the measured outcome branching ratio below the +2.0 V threshold is less readily explained. The toluene bond-breaking reactions studied here, desorption and switching, can be described by the theory of dynamics induced by electronic transitions (DIET). In a DIET process: (i) the injected electron is captured by an electronic state of the molecule causing a transition of the molecule from a neutral potential energy surface to an ionic-state potential surface; (ii) if the molecule is not at an equilibrium point of that excited state surface it will evolve along that potential causing a positional shift of the molecule; (iii) after a lifetime of typically a few femtoseconds the excited state decays (the electron transfers to the silicon substrate) leaving the now neutral molecule in a non-equilibrium position on the neutral potential energy surface - that is, the molecule will be a vibrationally excited neutral molecule; (iv) the molecule will then evolve in this vibrationally excited state and can, with a certain probability, lead to the reaction outcomes we observe[18].

Through the constrained choice of the molecular reaction outcomes and the picometre control of the injection location, we ensure that step (i), the initial excitation, is the same for all manipulated molecules. We previously showed that the lifetime of the excited state in step (ii) and hence the associated probability of manipulation can be regulated by the proximity of the STM tip[27]. The range of currents and injection bias voltages probed here result in a tip-sample separation that is consistent with the region where there is no observable quenching of the excited state lifetime[27]. Furthermore, we can discount any mechanical or electric-field manipulation by the STM tip since the lack of a current dependence at fixed voltage (Fig. 3) is also a lack of tip-height dependence. During these measurements the STM tip height varies by over ≈200 pm, while the branching ratio remains nearly constant over this range of heights.

To further probe the molecular reaction, we performed density functional theory (DFT) calculations to investigate the neutral ground state (neutral), the ionic resonance (anion), and the first excited state of the ionic resonance (anion*) of our system using the quantum chemical cluster models described in refs. 21,38. The optimised geometries (Supplementary Note 5) show that the reaction mechanism is mediated through the anion state. The effect of excitation into the anion* state for higher injection bias voltages was negligible ($\delta E = 0.01 - 0.04$ eV). The geometry of the anion* state was found to be between that of the neutral and anion states. In the anion state, the relaxed geometry of the molecule shows an elongation of the bond between the attaching Si adatom and the atom in the layer beneath it. This leads to a vertical displacement of the molecule away from the surface, as a precursor to the observed manipulation outcomes[26,38]. According to our simulations, only $0.1e$ of the additional negative charge remains on the molecule as calculated by an electrostatic potential-derived method. Most of the charge is transferred to one of the Si adatom bonds to a Si surface atom, leading to a low-frequency vibration of that bond and its subsequent breaking[38]. The maximal energy gained by the system during a single DIET jump cycle, $\Delta E_{\text{DIET}} = 0.2 - 0.4$ eV, from the anion* to the neutral state was

insufficient to induce manipulation. Therefore we conclude that the measured bias dependence of the branching ratio is not associated with excitation into one electronic state leading to one outcome and a different electronic state leading to another.

If the initial electronic excitation is consistent in position and state, and the excited state dynamics are unperturbed by the presence of the tip and, by extension, the resulting molecular dynamics are also unperturbed by the presence of the tip, it must be that it is purely the energy of the exciting electron that leads, through DIET, to the energy-dependent outcomes. Therefore, a monotonic increase of the branching ratio with injection energy implies a sensitive dependence of the evolution of the excited state, and the associated relaxation pathway of the molecule, on the energy of that initial excitation.

As with adsorption[39], for a molecule to transfer from chemisorbed to desorbed or switched, it must pass through a physisorbed state[25,26]. Therefore, we develop a simple model of that physisorbed state for the temperature dependence of the rate from the physisorbed state to re-attachment (switching) or to complete desorption. A full theoretical description and calculation are beyond this study, but we hope our model can provide enough physical insights to encourage more in-depth modelling. We had assumed that at threshold we would see only switching as this requires lower energy than complete desorption. However, we always find more desorption than switching: $k_d > k_s$ therefore $B > 1$.

We consider a model where following a DIET-like excitation and an ultrafast relaxation of the electron within the toluene LUMO state, the molecule is left in a thermally hot physisorbed state that can either: (1) chemisorb back to the surface with the same restatom but a new adatom site (switching), or (2) completely detach from the surface (desorption). If we take the ratio of these two outcomes and assume that our system is purely thermally driven from that physisorbed state, we arrive at a simple Arrhenius-type relationship of the form

$$B = \frac{k_d}{k_s} = \frac{A_d}{A_s} \exp\left(-\frac{\Delta E}{E_0 + \gamma[e(V - V_{\text{Th}}) + E_s]}\right). \quad (2)$$

Here, $A_d$ and $A_s$ are the respective Arrhenius pre-factors of desorption and switching and taking the experimental values reported in ref. 19 gives $A_d/A_s = 1.2 \times 10^3$. The excess energy of the injected electron is given by $V - V_{\text{Th}}$ where $V$ is the STM bias voltage and $V_{\text{Th}} = 1.4$ V is the voltage threshold for manipulation. The energy barrier to switching from the physisorbed state $E_s$ has previously been reported to be 0.3–0.4 eV from experiments done with benzene[40–42]. Experimental results for the energy barrier from the chemisorbed state to the physisorbed state $E_{\text{CP}}$ showed differences between toluene, benzene, and chlorobenzene in the range of only ≈ 50 meV. In our model, we chose to use $E_s = (0.35 \pm 0.05)$ eV. $\Delta E$ has previously been measured to be $(0.22 \pm 0.05)$ eV[19]. These values are in close agreement with the DFT-derived values presented in the Supplementary Note 6. $E_0$ and $\gamma$ are fitting parameters where $E_0$ is the base thermal energy of the physisorbed state, and $\gamma$ is a dimensionless parameter describing the efficiency of converting excess electron energy into thermal energy. Figure 5b shows the good fit of this thermal model to the measured branching ratio with $E_0 = (30.8 \pm 3.0)$ meV and $\gamma = (0.012 \pm 0.002)$. This corresponds to a base temperature of $(358 \pm 35)$ K (surprisingly close to the base room temperature of our STM and sample) and an increase of temperature of $(142 \pm 29)$ K per bias volt. The error bars presented correspond to one standard deviation of the fit to the model. Taking the extremal values for $E_s$ and $\Delta E$, the results fall within the statistical error bars.

In this model, the increase in the measured branching ratio would arise from the increase in the thermal energy of the physisorbed state at higher injection bias voltages. A similar fit using values informed by the DFT calculations in Supplementary Note 5, also results in a good fit to the experimental data, with a corresponding temperature increase

of 10 KV$^{-1}$ in the manipulation energy window with the difference due to the differing Arrhenius pre-factors, measured or calculated.

Gaebel et al.[39] showed that a chlorobenzene molecule chemisorbed on a Si(111)−7 × 7 surface and excited from its equilibrium geometry never gains the required energy for desorption through the anion resonance−even when the excited state lives for as long as 300 fs. According to our calculations, the geometries of a toluene molecule in the chemisorbed, transition, and physisorbed states are comparable to those of chlorobenzene (see Supplementary Note 5). Therefore toluene should exhibit similar dynamics to chlorobenzene. This implies that the extra thermal energy gained according to our model is key for stimulating the desorption process in ours and similar systems.

Crucially, our results indicate that it is possible to go beyond setting up the initial conditions of a charge-stimulated chemical reaction and instead gain control by using the whole range of the molecular dynamics at play. In the future, if we can map the electronic excitation of adsorbates to specific vibrational modes of the molecule[11,16,43], we could achieve direct selectivity over the reaction outcome. Developing new control protocols based on this notion could enable more precise programming of novel molecular machines[44].

## Methods

### Sample and tip preparation

Experiments were performed with an Omicron STM 1 operating at room temperature and a base pressure of ≈1 × 10$^{-10}$ mbar. Si(111) −7 × 7 samples, tungsten tips and toluene molecules were prepared following the procedures outlined in ref. 20. The Si(111)−7 × 7 surface reconstruction was obtained from pre-cut Si(111) samples (n-type, phosphorus-doped, 0.001−0.002 Ω cm) by repeated resistive heating to 1250°. Tungsten tips were etched in a 2 M NaOH solution and outgassed in vacuum to remove any tungsten-oxide[45]. Toluene was purified by the freeze-pump-thaw technique with liquid nitrogen and checked for purity with a quadrupole mass spectrometer. We chose toluene for this study because of its thermal stability at room temperature, its ease of STM-induced molecular desorption, and its lack of STM current-induced intramolecular bond breaking[26]. To prepare a partially toluene-covered surface (≈3 molecules per unit cell) the Si(111)−7 × 7 surface was dosed through a computer-controlled leak valve. Stability during the injection was ensured by a drift-compensation software which limited sample drift to between 100 fm s$^{-1}$ and 10 pm s$^{-1}$ in each of the $x$, $y$ and $z$ directions. All voltages are applied to the sample with the tip grounded through a Femto pre-amplifier.

### Automated local single-molecule experiments

To manipulate ≈120 individual molecules at each set of injection parameters (tunnel current and bias voltage) we used a home-made LabVIEW control programme running in combination with a bespoke MATLAB analysis suite[45]. Each automated experimental sequence involves the following steps: (1) LabVIEW takes a 25 nm × 25 nm overview image of the silicon surface; (2) MATLAB analyses the image to identify all atomic and molecular locations and randomly picks a user-specified number $n$ of injection sites. All selected molecules are located on top of faulted corner silicon adatoms, belonging to different unit cells; (3) LabVIEW moves the STM tip to the first set of $x$–$y$ coordinates specified by MATLAB. The STM takes a 3 nm × 3 nm image of the surface and performs a cross-correlation routine with the same region cropped out from the "big" overview image until it corrects for any drift that has occurred in the interim. (4) Once the correct injection location has been established, the STM takes a series of three 3 nm × 3 nm consecutive images, as described in Fig. 1 and the main text; (5) Steps (3) and (4) are repeated $n$ times for each set of injection co-ordinates; (6) Finally, the STM takes a 25 nm × 25

nm overview image of the silicon surface after all $n$ injection experiments have been performed. Each experimental sequence like this takes about 15−20 min and involves up to 20 individual current injections into single molecules. See ref. 46 for a time-lapse video of the automation procedure.

The automated LabVIEW virtual instrument for manipulation injections was altered to increase the sampling rate to 10,000 Hz and minimise wait times within the script to increase the passing of information from the Nanonis electronics to the computer to every 25 ms. The process of injection from ref. 27 was improved by altering step (ii) so the tip no longer initially approaches to 20 pA to avoid overshooting, but instead, the feedback loop is reduced following the tip's retraction by 1 nm from the surface. This allows the tip to approach the required injection parameters smoothly to start the injection. Every segment of 25 ms sampling time, the LabVIEW programme calculates the average $z$ height for that time segment and compares it to the average $z$ height for the very first 25 ms of the injection to find $\Delta z_M$ on Fig. 1e, the height of the tip retraction during a manipulation event. If this $\Delta z_M$ value is larger than the user-specified delta $z$ trigger, then the injection halts as a manipulation event has been detected. The current is then set to 5 pA while the feedback loop is turned off and the tip is retracted 1 nm away from the surface, and the current and voltage are re-set back to passive imaging parameters of +1 V and 100 pA as before.

### Density functional theory simulations

The quantum chemical cluster model geometry optimization calculations were performed using the B3LYP hybrid functional, employing the 6-31G* basis set and the Grimme D3 dispersion correction as implemented in Gaussian09 Revision D.01[47]. For the excited states, linear response time-dependent density functional theory with the same functional and basis set was used. The cluster model A of the Si(111)−7 × 7 surface comprises 17 silicon atoms and 26 saturating hydrogen atoms, while model B is slightly larger (21 silicon, 34 hydrogen atoms) as it contains all silicon atoms up to two bonds distance from the adatom-restatom pair that are responsible for the bonding to toluene. The nomenclature used to label atoms and methyl group substitution positions within the cluster is defined in Supplementary Note 5. In previous work, we showed that the clusters used in these calculations (A and B) are sufficient to describe the behaviours of the system and that they perform well in comparison to a much larger and more computationally expensive model (67 silicon, 54 hydrogen atoms)[38]. The neutral ground state (neutral), the first excited state of the negatively charged system (anion*) and its ground state (anion) were optimised for the A-PhCH3 and B-PhCH3 cluster models, as detailed in Supplementary Note 5. From the total energies of each state at the different optimised geometries, we estimated the maximal energy $\Delta E_{max}$ which can be gained during a DIET jump cycle.

The transition barrier calculations were done on the M062X-D3/defTZVP level of theory using Gaussian16 Revision C.01[48]. Two reaction paths were investigated, i.e. the ones to B-PhCH$_3$-R3-b and B-PhCH$_3$-R5-a, starting from the two most stable positions of the CH$_3$ group[21]. The two other possible paths, i.e. the ones to B-PhCH$_3$-R3-a and B-PhCH$_3$-R5-b (a and b label opposite direction for the displacement of the physisorbed molecule c.f. Suppl. Note 6 and ref. 39), were not considered because here the methyl group would have to pass the adatom. This leads to a large amplitude motion in the lateral direction and possibly to additional barriers, which complicate the identification of a reliable transition state.

## Data availability

The data that support the findings of this study are available from the University of Bath data archive[49] and from the corresponding author upon request. Source data are provided with this paper.

## Code availability

The Codes used for analysis of the data are available from GitHub (https://github.com/Bath-STM/branchRatio), from the University of Bath data archive[49], and from the corresponding author upon request. The code is provided under a GNU General Public License v.3.0.

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

## Acknowledgements

The authors thank Dr Victoria Scowcroft for discussions and advice on the fitting procedures. This work was supported by the Royal Society (No. RGS/R1/231369, K.R.R.), the Engineering and Physical Sciences Research Council (EPSRC) (No. EP/X031934/1, K.R.R. and EP/L015544/1, RMP), and by a University of Bath studentship (P.J.K.).

## Author contributions

R.M.P. and K.R.R. performed the experiments and the analysis. P.J.K. performed the data analysis, helped write the paper and interpret the results. T.K. provided theoretical support and performed the simulations. K.R.R. and P.A.S. designed the experiments and the analysis and wrote the manuscript.

## Competing interests

The authors declare no competing interests.
