## [Transparent Peer Review file · Nature Communications]

Measuring competing outcomes of a single-molecule reaction reveals classical Arrhenius chemical kinetics

Corresponding Author: Dr Kristina Rusimova

Version 0:

Reviewer comments:

Reviewer #1

(Remarks to the Author)

The manuscript entitled "Controlling Competing Outcomes of a Single-Molecule Reaction" by Rebecca M. Purkiss et al. presents a comprehensive investigation into the manipulation of toluene molecules adsorbed on a Si(111)-7x7 surface through atom manipulation events. The authors offer a robust technical framework and articulate the outcomes of their experiments with commendable clarity. The presentation of results, although intricate, is articulated in a manner that should be accessible to the diverse readership of Nature Communications.

A particularly noteworthy aspect of this work is the application of the classical Arrhenius equation to single-molecule on-surface manipulation events, where the tunneling current voltage serves as a substitute for temperature. This finding underscores a fascinating bridge between 19th-century foundational chemical kinetics and cutting-edge nanoscale science, rendering the study a potential cornerstone for educational purposes in undergraduate curricula.

Despite the significant contributions of this study, I have identified several areas where further clarification or additional investigation could enhance the manuscript:

1. The possibility of calculating intrinsic reaction coordinates for switching and desorption events merits exploration. It is conceivable that these processes may share a common intermediate state, suggesting a more nuanced reaction pathway than initially presented. Clarifying this aspect could provide deeper insights into the mechanistic underpinnings of these competing reactions.
2. The manuscript omits consideration of scenarios where the bond between Si and toluene is broken without subsequent displacement of the molecule on the surface, a condition referred to as a non-reactive event (III). Including an analysis of why such events were excluded from the statistical assessment could offer a more comprehensive understanding of the surface reactions under study.
3. The anomalously low manipulation rate for desorption observed at 1.8 V in Figure 4a diverges from the broader trend and lacks a thorough discussion. Addressing this discrepancy could strengthen the reliability of the findings and provide a clearer picture of the reaction dynamics.
4. The manuscript discusses a reaction driven by 0.5 electrons, as inferred from data fitting, which appears counterintuitive. An expanded discussion on the methodology and implications of this finding could clarify its relevance and validity within the context of the study.
5. I recommend revising the title of the manuscript to incorporate the applicability of the classical Arrhenius equation to nanoscale reactions. This adjustment would more accurately reflect the groundbreaking integration of traditional chemical kinetics with contemporary nanotechnology, underscoring the manuscript's innovative contribution to the field.

Minor Remarks:

1. The molecular illustration in Figure 1b resembles p-xylene, not toluene, which could lead to confusion. Ensuring the accuracy of such details is crucial for maintaining the scientific rigor of the manuscript.
2. The description of the negative ion resonance-induced manipulation technique in the text suggests a minor revision for clarity. Specifically, the term "preset tunneling parameters" might be more accurately expressed as "preset tunneling current parameters" to reflect the intended experimental conditions accurately.

In summary, this meticulously executed and presented study aligns well with the diverse readership of Nature Communications, showcasing a blend of fundamental chemical principles in advanced nanoscale experimentation. Consequently, I recommend its publication, contingent upon the resolution of the previously outlined comments.

Reviewer #2

(Remarks to the Author)

This work aims at achieving control over the outcomes of a chemical reaction by tip-manipulation events. The two outcomes are based on molecular desorption or molecular switching to a different site on the Si(111)-7x7 surface. Exceeding the excitation energy of the molecular electronic structure, a local thermal effect is generated and control over the two reaction outcomes is achieved. Although the topic is very attractive and this work provides information about molecular dynamics in chemical reactions, due to the current state-of-the-art the work by Purkiss, et al., does not represent a relevant advance suitable for publication in Nature Communications. Independently of the excitation energy, the desorption process is always favored, thus lacking control over the reaction outcomes. Control over the chemical reaction outcome by tip-position (Phys. Rev. Lett. 100, 136104 (2008)) or bias voltage-dependence reactions (e.g. J. Chem. Phys. 120, 5347–5352 (2004)) have already been demonstrated, some of them also studying the effect of electron exposure time (J. Am. Chem. Soc. 2011, 133, 26, 10066–10069). The main achievement of the manuscript is the time-dependent branching ratio, so this reviewer believes that from the perspective of novelty and significance, it does not meet the standards expected for publication in Nature Communications.

Additional comments:

1. The introduction needs to be elaborated, giving some context (e.g. Phys. Status Solidi B250, No. 9, 1671–1751 (2013)/ DOI10.1002/pssb.201248392) and explaining why this work means an advance in the field. Two sentences are repeated: “A scanning tunnelling microscope (STM) can initiate and probe these processes in atomically controlled environments. On-demand switching of weakly-bound, cryogenically-stabilised single molecules has been demonstrated in many systems [2–4]. Selectivity in the reaction outcome can be ensured by exciting different electronic states [5–7], changing the excitation location [8–10], or exciting different molecular vibrations [11, 12].”

“Examples of STM controlled chemical reactions have so far been demonstrated by changing the excitation mechanism, for example by selectively exciting different vibrations [12] or different ionic resonances of a molecule, or by changing the charge injection location within a molecule [3, 4, 9, 17, 18].”

2. Is the bifurcation taking place from the physisorbed state? Is at this point when the excess local heating process initiates a different reaction outcome?

3. The toluene-restatom sigma bond seems to be stronger than the toluene-adatom bond. This could be the reason why there are two different outcomes controlled by energy. Calculations about the energy barriers should be included in the main text and calculated charge transfer of toluene on Si(111)-7x7 surface should be provided.

4. The authors focused on the STM electronics limitations for determining the time-dependent process. However, the importance of the injection position, revealed in figure 3e, evidences a new parameter to be considered. Lateral drift compensation was performed in the experiments presented in Figure 2?

5. The branching ratio as a function of the energy of the injected electrons should be performed with position dependence. This trend could provide information related to the activation barriers of toluene-adatom and toluene-restatom.

Reviewer #3

(Remarks to the Author)

In this study the authors investigate with an STM at room temperature the electron-induced reaction (switching or desorption) of individual toluene molecules chemisorbed on Si(111) 7x7 surface. They report that the probability of the reaction outcome, desorption vs switching can be control with the bias voltage between the tip and the sample. With the support of DFT calculations, they explain this effect with a thermal type effect on an intermediate physisorbed state induced by the excess energy of the hot electrons injected in the molecule.

The authors have worked with this system since many years and have a really good knowledge and control of the induced chemical reaction they generate, with a well-defined and explained methodology. The analysis is carefully done and the model they suggest seems reasonable and well supported by their results (Experiment and DFT); and also, from my knowledge never reported before. Thus, I think that this work is on high quality and may interest the broad community of surface science.

However, before I can definitively recommend this work for publication in Nature communication, I have one main restraint to the form of actual version of the article and several questions/remarks for the authors. My main problem is that I found really hard to go through the whole article and get the main point of the work. I think this problem arises because the authors have worked and published since many years on this system, and that a lot of important information required to understand properly this work are disseminated in their numerous previous publications and often not properly integrated to this paper (meaning not possible to understand without reading the complete previous papers). This integrated the following points:

- The description of the studied system, a non STM specialist would have a hard time to understand how you can distinguish what is a toluene molecule on the surface (better resolved and larger area STM scan of your system in supplementary information may already help).
- The reaction mechanism with the DIET model and the role of the tip coupling to the excited state to understand the current dependence.
- DFT calculations: the influence of the different cluster sizes and the different methyl group positions.

d) The question of why is the desorption more efficient than the switch. This question which is not really relevant for the main point of the work stay pending the whole paper and may mislead the reader.

I would advise the author to try to integrate these points more carefully in their paper (or in the supplementary) in order that any reader which does not know their previous work can understand the main point of their work.

Additionally, I have the following questions/remarks:

1) The reaction path involved a metastable physisorbed state with a barrier of $\sim 0,3\text{eV}$ (not so small) to go back to the chemisorbed state. Should that not be possible to stabilize this intermediate state and experimentally observed it?

2) The control of the outcome involved a thermal excitation. Thus, changing the based temperature of the experiment should have an effect on the result of the model. I saw that the authors have already measured this system at 77K (citation 28). Have they results at this temperature which may confirm their model (and make it more robust).

3) For the fitting model, some fitting parameters are based on experimental values related to citation [29]. But the corresponding citation is "Time lapse of automatic STM manipulation [video]. <https://youtu.be/>" which does not seem the correct one.

4) The fit of the model is either based on experiment on benzene molecule (citation [29?-30-31]) or dependent of the DFT calculation. Considering possible differences in the considering energies between toluene and benzene and considering energy values obtained by DFT may be quite dependent of the DFT parameter choose (or size of the surface cluster). How robust is the model considering some possible input disparity (like ratio of E_s/E_d)?

5) In page 7 the authors write: "Both the rate of desorption and the branching ratio increase monotonically with the energy of the injected electrons."

If it's true for the branching ratio, it's clearly not for the rate, which clearly drops between 1.4 and 1.6V, and between 1.7 and 1.8V (More an oscillatory behavior). And these variations are quite outside the error bars shown in the graph. How the authors can explain this behavior? Is it reproducible on several set of data?

6) In page 4 the authors write: "(iv) after 2 ms of electron injection, the molecule reacts:". Should that not be 20ms when looking on the figure?

Version 1:

Reviewer comments:

Reviewer #1

(Remarks to the Author)

The authors have thoroughly addressed all of my previous concerns, providing satisfactory responses and making the necessary revisions to strengthen the manuscript. The improvements have significantly enhanced the clarity and impact of the work. I am now confident in the quality and scientific merit of this study and recommend it for publication in Nature Chemistry.

Reviewer #2

(Remarks to the Author)

The authors have addressed my previous concerns in their thorough revision. The new introduction and new figures help to understand the main message and emphasize the significance of the work. Additionally, I think that the new title is more accurate. I still believe that the novelty of the manuscript is not enough for nature communications standards, therefore I leave the decision to the editor criteria.

In case the manuscript gets accepted, some typos should be amended:

- Page 9: desorprtion for desorption
- Page 4: "The negative ion resonance-induced manipulation of the single toluene molecule in Fig. 1a ..." I think it corresponds to Fig. 1c
- In Fig 3c the molecule is a benzene instead of a toluene.
- Figures from appendix B seem to be mixed with the references.

Reviewer #3

(Remarks to the Author)

The authors answered to my questions and remarks, and the modifications clearly improve the quality of the paper. Therefore, I recommend the paper for publication in Nature communication.

We thank the reviewers for their detailed and helpful comments on our work which we believe have improved the manuscript significantly, hence the time taken to reply (and various life events in the team). We are very pleased to hear that the reviewers found our work “comprehensive”, “high quality”, “robust”, “never reported before”, and “a potential cornerstone for educational purposes in undergraduate curricula”. Below, we provide a point-by-point response to the reviewers’ comments.

Response to Reviewer #1:

1. The possibility of calculating intrinsic reaction coordinates for switching and desorption events merits exploration. It is conceivable that these processes may share a common intermediate state, suggesting a more nuanced reaction pathway than initially presented. Clarifying this aspect could provide deeper insights into the mechanistic underpinnings of these competing reactions.

This is true, the model is very simple and the transition state pathways warrant further exploration. However, we feel that a more detailed theoretical work is beyond the scope of the present paper. We note that the discussion in the paper does indeed involve a common intermediate state.

2. The manuscript omits consideration of scenarios where the bond between Si and toluene is broken without subsequent displacement of the molecule on the surface, a condition referred to as a non-reactive event (III). Including an analysis of why such events were excluded from the statistical assessment could offer a more comprehensive understanding of the surface reactions under study.

They are excluded as we cannot set a discriminator for this in our z-stop experiments – reattachment to the original adsorption site happens on a timescale much faster than the response time of the STM control electronics. Therefore, we cannot tell if the molecule has re-attached at the original spot. However, looking at the branching ratio between desorption and switching is enough to get a qualitative understanding of the excited state dynamics, without mapping all the possible outcomes with absolute rates. We have added the following text and additional Supplementary information to the manuscript:

“The scenarios where the molecule does not react or reattaches to its original adsorption site, following an excitation into the physisorbed state, do not change the overall behaviour of the measured outcome branching ratios but just the overall values. Therefore, they are deliberately excluded from our analysis of the reaction outcome branching ratio (see Supplementary section A.6 for a detailed reaction outcome probability tree).”

3. The anomalously low manipulation rate for desorption observed at 1.8 V in Figure 4a diverges from the broader trend and lacks a thorough discussion. Addressing this discrepancy could strengthen the reliability of the findings and provide a clearer picture of the reaction dynamics.

This is a very good point, and one we apologise for not clearly addressing in the manuscript in the first place. We have extended the experimental data to include voltages above +1.9 V and reanalysed all of our experiments. By setting the total number of manipulation events to account for the actual number of experiments attempted, we were able to extract a more correct value for each manipulation rate. In addition, it is worth pointing out that the data acquisition process involved randomising the injection energies, locations, and currents at which we inject each day, and acquiring the data across many months. This procedure should rule out any effects due to temperature variations in the room and should result in data acquired across a range of different tip states. However, the absolute manipulation probabilities are very sensitive to the tip states (as the tip state determines the fraction of

electrons captured by the molecule), as well as the very exact position of the tip on top of the molecule. One of the main strengths of the analysis that we showcase in this report is that the branching ratio normalises out these uncontrollable (for the moment) experimental parameters as Figure 3 d & e starkly shows.

Based on the additional data, we have added a new Figure 4 and amended the existing Figure 5 to include the now correct manipulation rates.

4. The manuscript discusses a reaction driven by 0.5 electrons, as inferred from data fitting, which appears counterintuitive. An expanded discussion on the methodology and implications of this finding could clarify its relevance and validity within the context of the study.

In a similar spirit to the previous comment, after taking into account in the analysis the exact number of manipulated molecules and those that remained unreacted, the number of electrons comes out to be (0.8 ± 0.1) for both desorption and switching. The obvious non-integer value is because we are fitting a straight line to the absolute rates of manipulation and these are modified to some extent by the tip-state and the very exact injection position. Therefore, the error bars which are based on the statistics are an underestimation and do not account for uncertainty introduced by variations in tip states, etc. From the branching ratio plot below (Fig. 3 b) we can see that the measurement is invariant even though the absolute rates are moving – again, we are normalising out the tip effects. We have added the following text:

“The number of unreacted molecules is included in our analysis of the probability of manipulation, but is explicitly excluded from branching ratio analysis.” and

“For desorption $n = (0.8 \pm 0.1)$ and for switching $n = (0.8 \pm 0.1)$ which suggests that both outcomes are driven by a one-electron process. Such non-integer results are typical for these experiments due to the sensitivity the raw probabilities can have on the precise location of the tip relative to the target or the tip state. For example, the pioneering early STM reports of hydrogen desorption from Si(100) gave $n = 15$ [28] or $n = 10$ [29]. However, a following report with noticeably larger current range, reported $n = 0.3, 1.3, 0.3$ depending on the exact experimental procedure, evidence of (on average) a 1-electron process. [30] Therefore, the absolute rates, k_d and k_s , will depend on the exact state of the STM tip, whereas the branching ratio of the rates will naturally lead to a normalization of these tip-state effects. Fig. 3b shows a near constant branching ratio over the probed tunnelling current range. This invariance with tunnelling current further shows that both outcomes are one-electron processes and that there is no tip-induced quenching of the excited state [27]. For multi-electron processes, e.g. dynamics induced by multiple electronic transitions (DIMET), the branching ratio can exhibit orders of magnitude difference when the molecule is excited with different tip states or with different currents [10, 17].”

5. I recommend revising the title of the manuscript to incorporate the applicability of the classical Arrhenius equation to nanoscale reactions. This adjustment would more accurately reflect the groundbreaking integration of traditional chemical kinetics with contemporary nanotechnology, underscoring the manuscript's innovative contribution to the field.

Title changed to “Measuring competing outcomes of a single-molecule reaction reveals classical Arrhenius chemical kinetics”.

Minor Remarks:

1. The molecular illustration in Figure 1b resembles p-xylene, not toluene, which could lead to confusion. Ensuring the accuracy of such details is crucial for maintaining the scientific rigor of the manuscript.

Figure fixed.

2. The description of the negative ion resonance-induced manipulation technique in the text suggests a minor revision for clarity. Specifically, the term "preset tunneling parameters" might be more accurately expressed as "preset tunneling current parameters" to reflect the intended experimental conditions accurately.

Text re-worded.

Response to Reviewer #2 (Remarks to the Author):

Independently of the excitation energy, the desorption process is always favored, thus lacking control over the reaction outcomes. Control over the chemical reaction outcome by tip-position (Phys. Rev. Lett. 100, 136104 (2008)) or bias voltage-dependence reactions (e.g. J. Chem. Phys. 120, 5347–5352 (2004)) have already been demonstrated, some of them also studying the effect of electron exposure time (J. Am. Chem. Soc. 2011, 133, 26, 10066–10069).

This is precisely the point – in all of these studies the control is achieved by changing the initial conditions of the excitation. The reaction is then allowed to evolve naturally, and the outcome is set by those initial conditions. Here we look at subtly altering the properties of the intermediate excited state of the system and thereby changing the reaction outcome probabilities. We keep both the initial conditions and the excitation process of our test system the same: same injection site location, same electronic state excitation. We have re-written the introduction to give a clearer statement of significance and comparison to previous works. We hope that the new title also reflects the significance of this work more accurately.

Additional comments:

1. The introduction needs to be elaborated, giving some context (e.g. Phys. Status Solidi B250, No. 9, 1671–1751 (2013)/ DOI10.1002/pssb.201248392) and explaining why this work means an advance in the field. Two sentences are repeated:

“A scanning tunnelling microscope (STM) can initiate and probe these processes in atomically controlled environments. On-demand switching of weakly-bound, cryogenically-stabilised single molecules has been demonstrated in many systems [2–4]. Selectivity in the reaction outcome can be ensured by exciting different electronic states [5–7], changing the excitation location [8–10], or exciting different molecular vibrations [11, 12].”

“Examples of STM controlled chemical reactions have so far been demonstrated by changing the excitation mechanism, for example by selectively exciting different vibrations [12] or different ionic resonances of a molecule, or by changing the charge injection location within a molecule [3, 4, 9, 17, 18].”

We have re-written the introduction.

2. Is the bifurcation taking place from the physisorbed state? Is at this point when the excess local heating process initiates a different reaction outcome?

The bifurcation takes place from the physisorbed state. We have amended the diagram in Figure 4 to represent our model more clearly. We have also highlighted this point better in the introduction and throughout the text.

3. The toluene-restatom sigma bond seems to be stronger than the toluene-adatom bond. This could be the reason why there are two different outcomes controlled by energy. Calculations about the energy barriers should be included in the main text and calculated charge transfer of toluene on Si(111)-7x7 surface should be provided.

The key here is that we assume the initial excited state is the same for all outcomes. What those outcomes are is governed by their individual reaction pathways – so, yes, some are more or less likely due to their inherent energy barriers, as we argue, (and pre-factors). The injected charge does not localise on one bond or the other, causing one bond or the other to break. Instead, the charge goes to the surface, causing reorganisation of surface atoms and the breaking of the Si adatom - Si dimer atom bond first. This leads to an elevation of the Si adatom (and the molecule attached to it), eventually promoting the molecule into the physisorbed state and resulting in the observed manipulation (Ref. 26 & 38). The thermally activated migration of the physisorbed molecule and its affinity for middle adatoms sites has also previously been reported (Ref. 25) and likely aids the switching route. In addition, eventually always both bonds should be broken because this is crucial for the rearomatization of the physisorbed molecule. We have added the following discussion to the manuscript:

“In the anion state, the relaxed geometry of the molecule shows an elongation of the bond between the attaching Si adatom and the atom in the layer beneath it. This leads to a vertical displacement of the molecule away from the surface, as a precursor to the observed manipulation outcomes [26, 38]. According to our simulations, only 0.1 e of the additional negative charge remains on the molecule as calculated by an electrostatic potential derived method. The rest of the charge is transferred to the Si adatom - Si dimer atom bond, leading to a low-frequency vibration of that bond and its subsequent breaking [38].”

4. The authors focused on the STM electronics limitations for determining the time-dependent process. However, the importance of the injection position, revealed in figure 3e, evidences a new parameter to be considered. Lateral drift compensation was performed in the experiments presented in Figure 2?

Lateral drift as well as vertical creep compensation were performed before each injection experiment in the paper, ensuring a drift < 3 pm/s, resulting in a maximum drift of 24 pm for our longest experiments (of 8 s). Before reporting a measurement, we also check the half-images taken immediately before and after the manipulation experiment to make sure that the tip has stayed on target. By comparison, the size of the molecule is on the order of hundreds of pm. We have amended the caption of Figure 2 to emphasise this point.

5. The branching ratio as a function of the energy of the injected electrons should be performed with position dependence. This trend could provide information related to the activation barriers of toluene-adatom and toluene-restatom.

This could be an interesting experiment to do. However, based on the discussion we added in response to Point 3, the fact that we do not observe a second reaction threshold in the energy dependence of the branching ratio presented in Figures 4 & 5, as well as the observation that desorption (which has a higher energy barrier) is always favoured over switching (which has the lower energy barrier), we believe that such experiments will not reveal much further information about the manipulation process. Instead, we performed further experiments above +1.9 V, which allow us to gain better understanding of the energy

dynamics of the injected electrons. We have now included an extra Figure 4 and amended the existing Figure 5 to include these results. We also added a discussion explaining the role of the electron energy dynamics on the measured reaction branching ratios.

Response to Reviewer #3:

a) The description of the studied system, a non STM specialist would have a hard time to understand how you can distinguish what is a toluene molecule on the surface (better resolved and larger area STM scan of your system in supplementary information may already help).

We have amended Figure 1 to include larger-scale images of the Si(111)-7x7 reconstruction both with and without adsorbate molecules.

b) The reaction mechanism with the DIET model and the role of the tip coupling to the excited state to understand the current dependence.

Please see response to Reviewer 1 Point 4.

c) DFT calculations: the influence of the different cluster sizes and the different methyl group positions.

Discussion based on previous work added to the supplementary text.

"In previous work, we showed that the clusters used in these calculations (A and B) are sufficient to describe the behaviours of the system and that they perform well in comparison to a much larger and more computationally expensive model (67 silicon, 54 hydrogen atoms) [38]."

"The investigations also showed that the transition energies between each of the three states (table B2) did not strongly depend on the functional group position (R3 or R5)."

d) The question of why is the desorption more efficient than the switch. This question which is not really relevant for the main point of the work stay pending the whole paper and may mislead the reader.

The question has now been removed.

Additionally, I have the following questions/remarks:

1) The reaction path involved a metastable physisorbed state with a barrier of $\sim 0,3\text{eV}$ (not so small) to go back to the chemisorbed state. Should that not be possible to stabilize this intermediate state and experimentally observe it?

At room temperature the lifetime of the physisorbed state of the molecule is much shorter than the temporal resolution of the STM ($\approx 1\text{ ms}$). Previous work states an estimated physisorption lifetime of $<1\ \mu\text{s}$ (Brown, Science 279 (1998)) based on an estimated Arrhenius prefactor of 10^{13} . Using the experimentally measured prefactor of 10^{17} by Lock et al. (ref. 18), this lifetime is even lower $\approx 50\text{ ps}$. It is possible to observe the physisorbed state of the molecule at low temperatures.

2) The control of the outcome involved a thermal excitation. Thus, changing the based temperature of the experiment should have an effect on the result of the model. I saw that the authors have already measured this system at 77 K (citation 28). Have they results at

this temperature which may confirm their model (and make it more robust).

This is a very interesting point and would be an excellent test of the proposed model. It would require redesigning the experiment to account for the longer lifetime of the physisorbed state and therefore would constitute a standalone investigation. Unfortunately, we no longer have access to a low temperature system.

3) For the fitting model, some fitting parameters are based on experimental values related to citation [29]. But the corresponding citation is “Time lapse of automatic STM manipulation [video]. <https://youtu.be/”>; which does not seem the correct one.

Citation fixed.

4) The fit of the model is either based on experiment on benzene molecule (citation [29?-30-31]) or dependent of the DFT calculation. Considering possible differences in the considering energies between toluene and benzene and considering energy values obtained by DFT may be quite dependent of the DFT parameter choose (or size of the surface cluster). How robust is the model considering some possible input disparity (like ratio of E_s/E_d)?

It is true that we use the experimentally measured number for the physisorbed potential well of a benzene molecule rather than that for a toluene molecule and that the difference between benzene and toluene may affect the exact magnitude of this number. Lock et al. (Ref. 19) measure the chemisorbed potential well depth for toluene, benzene, and chlorobenzene and see a difference of about 50 meV depending on the choice of molecule. If we consider the possibility of a similar variation in the depth of the physisorbed well, it would still be smaller than the variation between the experimentally found values which range from 0.3 eV to 0.4 eV (Ref. 40-42), and the DFT result of ≈ 0.3 eV (depending on the DFT model). The energy barriers calculated with DFT and presented in the Supplementary section match our experimental energy threshold for desorption of toluene, as well as the previously experimentally derived value of ΔE . We used the range of values for the physisorbed well depth discussed above to refit our model, and the changes in the reported value of the base temperature E_0 and the conversion constant γ were much smaller than the error bars cited in the paper.

We have changed the following paragraph to incorporate this discussion:

“The energy barrier to switching from the physisorbed state E_s has previously been reported to be 0.3–0.4 eV from experiments done with benzene [40–42]. Experimental results for the energy barrier from the chemisorbed state to the physisorbed state ECP showed differences between toluene, benzene, and chlorobenzene in the range of only ~ 50 meV. In our model we chose to use $E_s = (0.35 \pm 0.5)$ eV. ΔE has previously been measured to be (0.22 ± 0.05) eV [19]. These values are in close agreement with the DFT derived values presented in the Supplementary section B.2. E_0 and γ are fitting parameters where E_0 is the base thermal energy of the physisorbed state, and γ is a dimensionless parameter describing the efficiency of converting excess electron energy into thermal energy. Fig. 5b shows the good fit of this thermal model to the measured branching ratio with $E_0 = (30.8 \pm 3.0)$ meV and $\gamma = (0.012 \pm 0.002)$. This corresponds to a base temperature of (358 ± 35) K (surprisingly close to the base room temperature of our STM and sample) and an increase of temperature of (142 ± 29) K per bias volt. The error bars presented correspond to one standard deviation of the fit to the model. Taking the extremal values for E_s and ΔE , the results fall within the statistical error bars.”

5) In page 7 the authors write: “Both the rate of desorption and the branching ratio increase monotonically with the energy of the injected electrons.”

If it's true for the branching ratio, it's clearly not for the rate, which clearly drops between 1.4 and 1.6V, and between 1.7 and 1.8V (More an oscillatory behavior). And these variations are quite outside the error bars shown in the graph. How the authors can explain this behavior? Is it reproducible on several set of data?

Please see our response to Reviewer 1 Point 3.

6) In page 4 the authors write: "(iv) after 2 ms of electron injection, the molecule reacts:". Should that not be 20ms when looking on the figure?

Typo fixed.

We thank the reviewers for their positive comments. Below, we provide a point-by-point response to reviewer 2.

Reviewer #2 (Remarks to the Author):

- Page 9: desorprtion for desorption

Typo fixed.

- Page 4: "The negative ion resonance-induced manipulation of the single toluene molecule in Fig. 1a ..." I think it corresponds to Fig. 1c

Fixed.

- In Fig 3c the molecule is a benzene instead of a toluene.

Fixed.

- Figures from appendix B seem to be mixed with the references.

Supplementary material now moved to a separate file, which solves this problem.